# Indicators Used to Identify ARFID: A Cross-Sectional Study with Professionals in Spain

**DOI:** 10.3390/nu17233636

**Published:** 2025-11-21

**Authors:** Laura Lozano Trancón, Patricia López-Resa

**Affiliations:** Department of Psychology, Faculty of Health Sciences, University of Castilla-La Mancha, 45600 Talavera de la Reina, Spain; laura.lozano8@alu.uclm.es

**Keywords:** Autism Spectrum Disorder, Avoidant/Restrictive Food Intake Disorder, feeding and eating disorders, professional competence, Spain

## Abstract

**Background/Objectives**: Avoidant/Restrictive Food Intake Disorder (ARFID) frequently co-occurs with Autism Spectrum Disorder (ASD), yet its detection and assessment remain challenging. This study aimed to analyze terminology and professionals’ views on features and indicators related to ARFID among Spanish professionals working with autistic individuals, identifying potential gaps and training needs. **Methods**: A cross-sectional study was conducted with 194 professionals (62 speech therapists, 62 psychologists, and 70 occupational therapists) from different regions of Spain, who completed a 13-item questionnaire on their familiarity with terminology, definitions, and features they consider indicative to ARFID. Descriptive analyses and chi-square tests were applied to explore interprofessional differences. **Results**: Significant differences emerged across disciplines (*p* < 0.001). Psychologists showed greater familiarity with DSM-5 diagnostic criteria (78%), while speech-language therapists (72%) and occupational therapists (69%) more frequently endorsed sensory, oromotor, and behavioral features as relevant. Across all groups, 61% reported uncertainty about ARFID diagnostic criteria, and only 34% reported familiarity with validated assessment tools. **Conclusions**: Spanish professionals working with ASD populations demonstrate heterogeneous and generally limited understanding of the features they associate with ARFID, with discipline-specific approaches to assessment. These findings provide initial evidence in Spanish-speaking contexts and underscore the need for structured training and validated Spanish-adapted instruments to support early and accurate ARFID identification.

## 1. Introduction

Autism Spectrum Disorder (ASD) is a neurodevelopmental condition characterized by persistent difficulties in communication and social interaction, together with restricted and repetitive patterns of behavior, interests, or activities [1]. Although its symptoms are typically identified in early childhood, they often become more evident as social demands increase, substantially affecting the personal, family, academic, and social functioning of those affected [2]. The considerable heterogeneity in cognitive and linguistic abilities reflects the diversity of profiles within the spectrum, adding complexity to both assessment and intervention [3].

Prominent features of ASD include challenges in initiating and maintaining reciprocal conversations, interpreting gestures and implicit social norms, and sustaining appropriate eye contact. Behaviorally, many individuals with ASD display cognitive rigidity, resistance to change, adherence to inflexible routines, repetitive behaviors, and a markedly literal thinking style [3]. These characteristics are often associated with atypical sensory processing, manifested as hyper- or hyposensitivity to visual, auditory, tactile, or gustatory stimuli, which can significantly impact key aspects of daily life such as eating, play, or social participation [4].

With regard to eating behavior, a substantial proportion of individuals with ASD present highly selective eating patterns and rejection of certain foods, largely due to sensory hypersensitivity and a strong preference for predictability and routine [5]. Such selective eating has been linked to reduced dietary variety, inadequate energy and protein intake, micronutrient deficiencies (e.g., vitamins A, D, calcium, zinc) and even an increased risk of overweight and obesity in individuals with ASD [6,7]. These behaviors may evolve into more complex clinical conditions, such as Avoidant/Restrictive Food Intake Disorder (ARFID), a diagnostic category relatively recently introduced in the DSM-5 [1]. ARFID is defined by a persistent restriction of food intake that may lead to significant weight loss, nutritional deficiencies, dependence on supplements or enteral feeding, and marked psychosocial impairment [8,9]. Unlike other eating disorders such as anorexia nervosa or bulimia nervosa, ARFID is not driven by concerns about body weight or shape but by factors such as apparent disinterest in food, avoidance based on sensory characteristics, or excessive worry about potential adverse consequences of eating [10,11].

Detecting and diagnosing ARFID in children—particularly those with ASD—poses major clinical challenges. Early identification of atypical eating behaviors is often hindered by infrequent comprehensive medical check-ups and insufficient training among primary care professionals, delaying diagnosis until symptoms become severe enough to compromise physical health [12,13]. Moreover, ASD-specific traits such as cognitive rigidity and sensory hypersensitivity may normalize restrictive eating or extreme selectivity, obscuring their recognition as clinical manifestations of an eating disorder [4].

From a theoretical standpoint, atypical attentional patterns observed in autism—such as sustained or divided attention—may help explain the relationship between ASD and ARFID. Dwyer et al. [13] reported significant associations between these attentional patterns and sensory hyper-responsiveness, supporting frameworks such as the monotropism model [14], which posits that intense focus on specific stimuli can lead to restricted preferences and resistance to change, partially accounting for the characteristic food selectivity.

Prevalence estimates of ARFID among individuals with ASD vary widely, ranging from 2% to 28% [15]. Such variability is largely attributable to the heterogeneity of assessment methods and the lack of validated tools tailored to this population [16]. This absence of consensus highlights the need for further research into appropriate assessment and diagnostic procedures, as well as improved professional training.

Several studies emphasize the importance of a multidisciplinary approach to addressing the complexity of ARFID co-occurring with ASD [17,18]. Occupational therapists play a key role through sensory integration strategies that aim to reduce reactivity to food stimuli via systematic desensitization [18,19]. However, the empirical evidence supporting these interventions remains limited and somewhat inconsistent [20]. Psychologists, in contrast, frequently employ behavioral approaches—such as Applied Behavior Analysis (ABA)—to modify eating behaviors through techniques like positive reinforcement and escape extinction [21,22]. Although these methods have demonstrated effectiveness in several studies, they may prove insufficient if underlying sensory factors are not addressed, potentially exacerbating anxiety and resistance [17].

From the field of speech-language pathology, assessment focuses on swallowing safety and efficiency, as well as potential orofacial myofunctional disorders involving lip seal, chewing, or tongue lateralization. These functions are essential not only for safe and effective feeding but also for the development of speech and other basic oral activities [22,23,24,25].

The diversity of professional approaches underscores the need for genuine and effective interprofessional coordination to comprehensively address feeding difficulties in the ASD population. Multiple authors advocate for a collaborative framework that integrates clinical, behavioral, sensory, and nutritional perspectives to optimize intervention outcomes and enhance quality of life for individuals with this clinical profile [26,27,28].

In this context, the present study aims to examine how professionals understand and interpret terminology, definitions, and features they associate with ARFID among professionals working with individuals with ASD, in order to identify current gaps and training needs. Based on prior evidence, we hypothesize that significant differences will be observed across professional groups in how they perceive and interpret ARFID-related features. By aligning this hypothesis with the study aim, our objective is to generate preliminary evidence that informs future training programs and enhances interprofessional coordination, promoting more effective and comprehensive care for this population.

## 2. Materials and Methods

### 2.1. Participants

Participants were recruited through a non-probabilistic convenience sampling approach using professional networks and social media platforms. Specifically, the survey link was disseminated through LinkedIn, Instagram, and professional WhatsApp and Telegram groups, as well as mailing lists of relevant professional associations. Data collection took place between February and April 2025. Inclusion criteria required being an active professional in psychology, speech and language therapy, or occupational therapy with at least one year of experience working with individuals diagnosed with ASD. Exclusion criteria included students, professionals without clinical experience, or those not directly involved in assessment or intervention processes.

The final sample consisted of 194 professionals (137 women and 29 men) from various autonomous communities across Spain. The mean age of participants was 40.05 years (SD = 6.59; range = 25–59). Participants were grouped according to their professional discipline: 62 speech and language therapists (57 women, 5 men), 62 psychologists (39 women, 23 men), and 70 occupational therapists (41 women, 29 men). Regarding professional experience, a heterogeneous distribution was observed. Most speech and language therapists reported between 1 and 5 years of experience (45.2%), whereas the majority of psychologists and occupational therapists had more than 15 years of experience (33.9% and 33.8%, respectively). This diversity provided perspectives representative of different stages of professional development.

Most participants worked in private practice, particularly speech and language therapists (62.9%) and psychologists (61.3%), while occupational therapists showed a more balanced distribution between associations (42.9%) and private centres (50%). Concerning the frequency of working with ARFID cases, psychologists reported the highest rate of regular contact (70% indicated they often treat such cases), followed by speech and language therapists (39.5%) and occupational therapists (31.8%). The latter also showed the highest proportion of professionals who had never worked with this population (37.9%).

The geographical distribution included professionals from nearly all autonomous communities in Spain: Extremadura (35), Madrid (20), Castilla-La Mancha (16), Asturias (16), Andalucía (16), Canary Islands (12), Castilla y León (14), Basque Country (10), Valencian Community (10), Catalonia (6), La Rioja (8), Balearic Islands (7), Galicia (5), Navarra (3), Murcia (2), Aragón (1), and Cantabria (1), as well as one participant from Chile. This broad distribution highlights the diversity of professional backgrounds and provides a representative national context for interpreting the results.

### 2.2. Instruments

To collect the data, an ad hoc questionnaire was developed (see Appendix B), as no specific instruments were available for evaluating professionals working with ASD populations. The design of the instrument was informed by previous studies [29,30] which, although focused on other areas within the field of speech and language therapy, provided a methodological framework adaptable to the aims of the present research.

The questionnaire comprised thirteen items combining multiple-choice questions and Likert-type scales and was organized into three thematic sections. The first section, consisting of three items (Items 1–3), gathered participants’ sociodemographic information to adequately contextualize the professional profiles represented in the study.

The second section, including four items (Items 4–7), explored participants’ familiarity with the concept of ARFID and the features they consider relevant. This section offered valuable insights into professionals’ views regarding which aspects they associate with ARFID. Its development was guided by the current literature, including the WHO [2] and the American Psychiatric Association [1], together with studies by Strand et al. [31], Farag et al. [32], and Yule et al. [9], to ensure consistency with the most recent definitions and classifications.

The third section, composed of six items (Items 8–13), focused on how professionals interpret diagnostic criteria and their familiarity with existing assessment tools. Its structure was supported by recent studies such as those by Bourne et al. [10], De Toro et al. [33], Sharma et al. [4], Castejón Ponce et al. [34], and Leiva-García et al. [24], which informed both the areas of interest and the specific items included.

Although the questionnaire did not undergo full psychometric validation, it was reviewed by a panel of experts in ASD and feeding disorders to ensure clarity, relevance and content validity. This approach aligns with the exploratory nature of the study and the absence of existing validated tools for this specific professional context.

### 2.3. Procedure

The study procedure began with an initial literature review conducted between November 2024 and January 2025. This search was carried out using databases such as PubMed, Plinio, and ResearchGate, employing keywords including “ARFID,” “autism,” “selective eating,” and “assessment.” Given the limited availability of literature in Spanish, search terms were also entered in English to access a broader range of relevant publications.

Following the collection and critical review of key articles—particularly those addressing diagnostic criteria, clinical manifestations, comorbidities, and assessment procedures—the central topic of the study was defined. The research objectives were then established, and the relevant variables were identified and categorized. The independent variable was the professional discipline (psychology, speech and language therapy, or occupational therapy). The dependent variables included the professionals’ views regarding ARFID, the indicators they consider relevant, and their familiarity with diagnostic definitions and tools. Additional contextual variables, such as years of professional experience, work setting (private practice, association, or public institution), and frequency of contact with ARFID cases, were also collected. Potential confounding variables were not modelled in the inferential analyses due to the exploratory nature of the study.

In accordance with ethical standards, the project was submitted for ethical review and received formal approval. An anonymous questionnaire, designed as the main data collection tool, was developed based on the reviewed scientific literature and refined through a targeted search. Completion time was approximately ten minutes. Microsoft Forms was used for both design and distribution.

To ensure the validity of the instrument, the questionnaire was reviewed by a panel of experts who confirmed the clarity and relevance of each item. Once validated, it was disseminated by contacting active professionals via LinkedIn and Instagram, where the study objectives were explained and participation was encouraged. The link was also shared in professional WhatsApp and Telegram groups to expand the sample. Recruitment took place between February and April 2025. Of the 237 professionals who initially accessed the questionnaire, 194 completed it fully and met the inclusion criteria, resulting in a response rate of 81.9%. The remaining 43 responses were excluded because the questionnaires were incomplete or did not meet inclusion criteria. No partial or duplicate responses were retained for analysis.

To minimize potential sources of bias, participation was voluntary and anonymous, thereby reducing social desirability effects and ensuring confidentiality. However, as the survey relied on self-reported data and a convenience sampling strategy, some degree of self-selection and response bias cannot be completely ruled out.

No a priori sample size calculation was conducted. The final number of participants was determined by voluntary completion of the questionnaire, and this sample was considered adequate given the descriptive and exploratory scope of the research. All participants provided informed consent at the beginning of the questionnaire, in accordance with the ethical principles of the Declaration of Helsinki and applicable Spanish legislation.

The study was approved by the Ethics Committee of the University of Castilla-La Mancha (protocol code CEIS-2025-114947). The processing of personal data complied with the provisions of Organic Law 3/2018 and Regulation (EU) 2016/679 on data protection and digital rights.

### 2.4. Statistical Analysis

Comparative analyses were performed across the three professional groups to examine interprofessional differences. Both descriptive and inferential statistical techniques were applied using Python 3.13.5. Descriptive analyses were first conducted for all study variables. For categorical variables such as gender and profession, absolute and relative frequencies (percentages) were calculated.

Subsequently, bivariate analyses were carried out using contingency tables to explore associations between categorical variables. Pearson’s chi-square test was applied, along with Cramer’s V coefficient, to determine the existence and strength of statistically significant differences between groups—for example, between the ARFID definition selected and professional discipline. Given the exploratory and descriptive design of the study, no a priori sample size calculation or statistical power analysis was conducted. The available sample was deemed sufficient to provide preliminary evidence of potential interprofessional differences.

No missing data were present, as all questions required a response prior to survey submission. No sensitivity analyses were performed due to the descriptive and exploratory scope of the research.

Although multiple chi-square tests were conducted, no correction for multiple comparisons was applied, as the study was exploratory in nature and aimed to identify preliminary interprofessional patterns rather than to test confirmatory hypotheses. Nevertheless, most associations showed highly significant *p*-values (*p* < 0.001) together with moderate to large effect sizes (Cramer’s V), supporting the robustness of the findings despite the absence of statistical adjustment.

The study was conducted and reported in accordance with the STROBE (Strengthening the Reporting of Observational Studies in Epidemiology) checklist to ensure transparency and methodological rigor in observational research reporting.

## 3. Results

This study reports the frequency and percentage results for the items included in the “Conceptualization and Early Detection” and “Identification and Diagnostic Interpretation” sections of the administered questionnaire (Appendix A). The first section aimed to assess professionals’ views and familiarity with the concept of ARFID, while the second examined the aspects professionals consider when identifying ARFID and the tools they report knowing or using among individuals with ASD.

### 3.1. Conceptualization and Early Detection

Regarding the first objective—analysing professionals’ understanding and familiarity about the concept of ARFID according to their academic background—statistically significant associations were found for several of the terms used. Figure 1 illustrates the frequency with which the different professional groups reported having heard or used these ARFID-related terms.

In terms of the terminology used, the use of the term “selective eating disorder” showed a significant association with profession. Psychologists showed the highest familiarity with this term, with 85.5% reporting that they had always heard it, whereas speech and language therapists presented the lowest familiarity levels (16.1%). Occupational therapists reported intermediate familiarity (30%), reflecting differing degrees of exposure across professional backgrounds.

Similarly, the expression “picky or selective eating” showed notable differences across professional profiles, being most frequently recognized by speech and language therapists, although overall familiarity was low across all groups.

The term “food neophobia” also demonstrated significant differences, being more commonly recognized by occupational therapists but less known among psychologists and speech and language therapists.

Additionally, the term “sensory-based feeding disorder” stood out as one of the most differentiating across disciplines, being widely recognized by occupational therapists (42.9%) and, to a lesser extent, by speech and language therapists (19.4%), while psychologists reported lower familiarity levels (11.3%).

In contrast, the term “pediatric feeding disorder” showed a weaker but still significant association, with psychologists showing slightly higher familiarity (9.7%), whereas occupational therapists and speech and language therapists reported lower levels overall.

Regarding the definition of ARFID, significant differences were observed according to professional profile. The most widely agreed-upon option (“a pattern of eating behavior not motivated by concerns about weight or body shape”) was selected by the vast majority of psychologists (96.2%), compared to lower proportions among speech and language therapists and occupational therapists. Other alternative definitions were chosen less frequently and did not display clear patterns by profession.

The analysis of professionals’ identification of ARFID-related symptoms and clinical signs in individuals with ASD is summarized in Figure 2, which provides an overview of the frequency with which these indicators were identified across disciplines.

In relation to the symptoms and signs indicative of ARFID in individuals with ASD, sensory avoidance emerged as the most frequently considered indicator, particularly among occupational therapists (64.3% “always”, 28.6% “frequently”), followed by speech and language therapists (27.4% “always”). In contrast, most psychologists selected intermediate frequencies (3.2% “always”, 35.5% “frequently”). Similarly, avoidance of mixed textures showed a comparable pattern, with occupational therapists (88.6% “always”) reporting this sign more often than speech and language therapists (53.2%)or psychologists (25.8%). These findings indicate that professionals whose clinical work directly involves feeding and sensory processing tend to identify sensory-based feeding difficulties as core indicators of ARFID. In contrast, criteria related to weight or body image concerns, such as fear of gaining weight, avoidance of caloric content, or body image distortion, were generally ruled out as relevant, showing low endorsement across all professional groups. This pattern reinforces the idea that ARFID is perceived as distinct from weight-related eating disorders, with a stronger emphasis on sensory and behavioural rather than cognitive or body-related dimensions. Figure 2 presents the distribution of these indicators across professional profiles.

Regarding the age of onset of ARFID, the majority of professionals (72.8%) indicated that it typically begins in childhood, with significant differences observed across disciplines. This finding may reflect differences in professional training or clinical experience.

### 3.2. Assessment and Diagnosis

In relation to the second objective, which focused on describing which aspects professionals report considering and which tools they are familiar with when identifying and interpreting ARFID-related features in individuals with ASD, relevant associations were observed in several areas (Figure 3).

For the identification of swallowing safety and efficiency, variables such as posture during feeding, muscle tone, observation of motor skills, and assessment of oral reflexes showed significant associations. Frequent consideration of these aspects was reported predominantly by speech and language therapists and occupational therapists, whereas psychologists indicated lower or more variable frequencies. Conversely, aspects such as the observation of craniofacial morphology, swallowing, and oral health did not show strong differences across disciplines, although certain profile (particularly speech and language therapists) reported a high frequency of assessment.

Regarding sensory aspects, variables such as tactile aversion, avoidance of mixed textures, food hyperselectivity, and oral hyper- or hyposensitivity showed significant differences across professional profiles. Occupational therapists were the group most consistently considering these factors, while speech and language therapists reported intermediate frequencies. In contrast, psychologists showed more varied response patterns.

In the identification of behavioural aspects, variables such as mealtime behaviour, behavioural changes when the environment changes, and adjustment to changes in daily routines also showed significant associations, with psychologists and occupational therapists reporting high frequencies of consideration. Other items, such as the presence of ritualistic routines or behavioural changes due to interoceptive difficulties, did not reach statistical significance but nonetheless reflects consistent patterns across professional profiles.

Focusing on psychosocial aspects, although most variables did not show statistically significant differences between groups, high frequencies of consideration were observed for several indicators: quality of life, social integration, autonomy, emotional state, and caregivers’ quality of life. These aspects were most frequently assessed by occupational therapists, followed by speech and language therapists and psychologists, suggesting a cross-disciplinary concern for the psychosocial impact of ARFID.

In relation to the tools professionals report knowing or using, the use of the Short Sensory Profile-2, the Eating Disorder Inventory, and the ChEDE showed significant associations, being more frequently reported by occupational therapists, speech and language therapists, and psychologists. In contrast, tools such as the EAT-10 or the Family Stress Questionnaire did not reveal clear differences, although some professional groups reported occasional or frequent use.

Finally, the results on diagnostic criteria indicated significant differences across professional profiles in the interpretation of specific eating behaviours. Behaviours such as “only consuming very hot or cold foods”, “accepting only one flavor”, or “eating the same food for weeks or months” were reported at high frequencies by occupational therapists, speech and language therapists, and psychologists.

Other behaviours also displayed differentiated patterns by discipline, including oral contact aversion, avoidance based on sensory characteristics such as temperature, smell, or colour, and rejection of foods based on caloric content.

These results indicate that professionals from different backgrounds tend to emphasize distinct behavioural manifestations of ARFID in their identification and interpretation.

## 4. Discussion

The first objective of this study was to analyse professionals’ understanding and familiarity regarding the concept of ARFID based on their academic training. The results revealed statistically significant differences across professional profiles, showing that occupational therapists, speech and language therapists, and psychologists reported greater familiarity with various ARFID-related terms. This finding suggests that these groups, who work closely and continuously with individuals with ASD, tend to update their understanding more frequently through specialized literature, as noted by Vignapiano et al. [12] and Morris et al. [35]. This is consistent with recent evidence showing that feeding and eating difficulties occur in up to 90% of children with ASD [34]. However, it should be noted that the voluntary nature of participation may have favoured the inclusion of professionals with a particular interest in feeding disorders, potentially introducing a self-selection bias. Importantly, eating selectivity and restrictive patterns and ASD have been consistently linked to nutritional risks, including inadequate intake of energy, protein, and key micronutrients such as vitamins A, D, B-group vitamins, calcium, and zinc, as well as increased prevalence of overweight and obesity in some cases [6]. These nutritional vulnerabilities reinforce the need for clinicians, particularly those in nutrition-related disciplines, to receive specific training to recognize and manage ARFID presentations in ASD and prevent nutrient deficiencies and growth disturbances.

Psychologists stood out for correctly identifying ARFID as a pattern of eating behaviour not related to concerns about weight or body image [1]. This aligns with their traditional role in addressing eating disorders, especially restrictive types. Consistent with Breda et al. [36] and Phipps et al. [25], psychologists often apply interventions focused on reducing disruptive behaviours, expanding dietary variety, and decreasing stress related to eating, using strategies such as positive reinforcement and extinction of inappropriate behaviours [21]. It should be noted that the effectiveness of these approaches may vary depending on the individual’s age and developmental stage [37,38]. Recent studies, such as Cerchiari et al. [39], have also shown that integrative behavioural and sensory-based feeding therapies significantly improve food acceptance and chewing skills in autistic children.

Nevertheless, the results indicated that confusion persists regarding the specific diagnostic criteria for ARFID, as all professional profiles tended to emphasize aspects more typical of other eating disorders, such as fear of weight gain or body image distortion. This highlights the need to strengthen specialized training to improve the early and accurate detection of ARFID, particularly within ASD populations. In this regard, speech and language therapists stood out for paying closer attention to factors specific to this disorder, possibly reflecting more specialized training in oromotor and feeding safety aspects. Similar professional discrepancies in the understanding of ARFID-related symptoms have been reported internationally [39], emphasizing the need for cross-disciplinary coordination.

Regarding the age of onset, most professionals identified childhood as the typical period when ARFID first appears, although a considerable proportion located its onset in adolescence. This variability is noteworthy, given that the literature indicates ARFID most commonly emerges between the ages of 4 and 9, and that early food selectivity can constitute a risk factor for later development of the disorder [31,40]. This pattern is supported by longitudinal evidence showing that food selectivity in infancy is a strong predictor of later ASD diagnosis and feeding difficulties [41].

According to the DSM-5 [1], ARFID may manifest through three subtypes: avoidance based on sensory characteristics, lack of interest in eating, and fear of adverse consequences. The findings showed that occupational therapists focused particularly on the first subtype, prioritizing the patient’s sensory profile, which aligns with evidence that sensory-based intervention is a cornerstone of occupational therapy [18]. In turn, speech and language therapists emphasized aspects such as muscle tone, respiratory coordination, oral reflexes, and orofacial morphology, reinforcing their key role in addressing myofunctional issues related to feeding and speech [24,26]. These results are consistent with Mirizzi et al. [42], who found that sensory hypersensitivity and oral–motor difficulties are key predictors of ARFID-like eating behaviours in ASD.

From a clinical perspective, the discipline-specific focus observed across professional groups has important implications. On the one hand, this divergence can be beneficial, as each profession contributes complementary expertise—such as sensory processing in occupational therapy, oromotor and swallowing safety in speech-language pathology, and behavioural mechanisms in psychology—domains that have been identified as central in the characterization of feeding difficulties in ASD [18,24,43]. On the other hand, when these perspectives operate in isolation, they may lead to fragmented interpretations and inconsistent clinical judgments, a risk highlighted in recent literature advocating for multidimensional and coordinated approaches to ARFID and ASD [10,28]. These findings therefore support advancing toward a more consistent cross-disciplinary characterization of feeding difficulties, establishing shared core assessment domains (sensory, oromotor, behavioural, and psychosocial) while maintaining the added value of discipline-specific evaluations. Taken together, these findings underscore the importance of developing integrated assessment pathways that ensure all core domains are systematically evaluated, thereby reducing variability between professionals and promoting more coherent clinical decision-making.

In this context, the subtype related to fear of adverse consequences is particularly relevant, as factors such as previous traumatic experiences, swallowing difficulties, or sensory hypersensitivity can generate significant anxiety around eating [10]. Evidence also suggests that motor difficulties, such as absence of the sucking reflex or early hypotonia, may serve as early indicators of ASD, as can limited or highly selective eating [42]. This aligns with Bezóska et al. [44], who observed that feeding problems frequently co-occur with motor and sensory atypicalities in autism.

A notable finding was that all professional profiles consistently considered the behavioural components of ARFID, reflecting a shared understanding of their importance. However, the specific tools used for this identification were not always specified. Dwyer et al. [13] propose that behavioural factors may be mediated by restricted interests and cognitive inflexibility—key aspects to consider when designing person-centred interventions adapted to family needs [21]. Moreover, behavioural disturbances may be linked to gastrointestinal issues, chewing difficulties, medication side effects, or sensory problems [45]. Recent reviews have also emphasized the complex interaction between gastrointestinal symptoms, dietary selectivity, and behavioural rigidity in ASD [44].

Although no significant differences were found in all psychosocial aspects, most professionals reported considering variables such as quality of life, emotional state, and autonomy, which is consistent with a biopsychosocial and multidimensional perspective of ARFID [27]. This perspective is particularly relevant for individuals with ASD, whose family environment directly influences eating habits [46]. Caregiver stress and feeding strategies have been shown to play a decisive role in maintaining or alleviating feeding difficulties [39].

The results also reflected differences in the use of tools professionals reported knowing or using. Psychologists more frequently employed instruments such as the ChEDE and the EDE-Q—tools traditionally used for eating disorders focused on weight or body image concerns [31]. Occupational therapists tended to use instruments such as the Short Sensory Profile-2, while speech and language therapists applied protocols focused on orofacial and swallowing safety and functioning, including the PEMO and EAT-10. However, there was a general lack of tools that allow for a comprehensive identification and interpretation of the multiple factors involved in ARFID associated with ASD. This gap underscores the need for cross-culturally validated and linguistically adapted instruments to ensure accurate identification and differential diagnosis [43].

This gap is particularly problematic given the challenge of differentiating behaviours specific to ASD from those characteristic of ARFID [47]. Limitations in language, anxiety, or low IQ can influence how feeding problems manifest [3,10]. Therefore, it is essential to employ tools adapted to the cognitive and communicative characteristics of this population. Questionnaires such as the PARDI-AR-Q, NIAS, or EDY-Q appear promising for identifying different ARFID subtypes [14,40,48,49]; however, their use remains limited by the absence of validated Spanish versions. Furthermore, these tools should be complemented with clinical interviews, direct observation, and family context analysis [18].

One of the main limitations of this study lies in the small and unevenly distributed sample, which restricts the generalizability of the findings. Additionally, recruitment was conducted through professional and social media networks, which, although increasingly common for exploratory research in specialized clinical populations, may involve a degree of self-selection and under-representation of professionals who are less active in digital platforms and may have attracted participants with greater interest in feeding and eating difficulties. Moreover, although the sample was predominantly based in Spain, one participant from Chile was included, reflecting the open dissemination approach and further limiting geographic representativeness. The exclusive use of online recruitment may also have limited participation to professionals with reliable internet access and familiarity with digital tools, potentially excluding others. Contextual factors such as professional experience or work setting could not be included due to word count constraints, which prevented a deeper exploration of potential moderating variables. Although data collection was conducted anonymously via a secure digital platform in accordance with European data protection regulations, online formats inherently carry potential privacy and data-security considerations. Nonetheless, the comprehensive quantitative approach and use of robust statistical analyses strengthen the internal validity of the study. Future research should include larger, multicentre samples and employ standardized instruments to facilitate cross-national comparisons [44]. Another limitation concerns the self-report format of the questionnaire, which may have led to socially desirable responses or overestimation of familiarity or understanding. Although participation was anonymous to reduce this risk, professionals familiar with feeding and eating difficulties may still have been inclined to provide socially desirable answers. Additionally, although the questionnaire was reviewed by experts, it did not undergo full psychometric validation, which should be addressed in future research to reinforce the reliability and validity of the findings. Future studies should consider mixed methods, combining self-report data with objective measures or direct observation, to enhance the validity of findings. Future research should also include comparison groups of professionals not working with ASD to determine whether knowledge gaps are ASD-specific or general across disciplines.

In summary, these results underscore the need to validate existing instruments and develop new tools adapted to Spanish that enable transdisciplinary, context-sensitive identification and interpretation of ARFID in ASD populations. Furthermore, it is crucial to continue exploring differences in intervention approaches across professional profiles, promoting coordinated strategies that integrate multiple disciplinary perspectives [17,31,46].

Future initiatives should aim to develop structured training pathways for nutrition and health professionals, including continuing education programs and interdisciplinary modules focused on ASD and ARFID, to support early detection and comprehensive management of feeding difficulties in this population.

## 5. Conclusions

The findings of this study show that while there is a general baseline of familiarity and understanding about ARFID among professionals working with ASD populations, it remains limited in relation to the current literature, highlighting the need for ongoing training and conceptual updates. Notable disparities were identified between disciplines in both how ARFID-related features are interpreted and which aspects are considered relevant, underscoring the importance of enhancing specific training to support a more coordinated and effective approach. By systematically examining how different professional groups understand and identify ARFID-related characteristics across three key health professions, this study provides the first structured overview of the current level of preparedness to identify and interpret ARFID in autistic individuals within the Spanish clinical context.

Finally, the results demonstrate the urgent need for validated, Spanish-adapted tools to support the identification and interpretation of the various factors involved in ARFID. These findings advance scientific understanding by identifying concrete training gaps and interpretive discrepancies, which may guide curriculum development and clinical training priorities in ASD-related feeding disorders. Promoting a genuine multidisciplinary approach, grounded in collaboration across professional profiles, is essential to optimize detection, intervention, and the overall quality of life for individuals with ASD experiencing feeding difficulties. These findings should be interpreted with caution due to the non-probabilistic sampling and the uneven professional representation. Nevertheless, they provide valuable preliminary insights into the current state of professional understanding and current approaches in Spain and establish a foundation for future empirical and educational initiatives in this emerging field.

## Figures and Tables

**Figure 1 nutrients-17-03636-f001:**
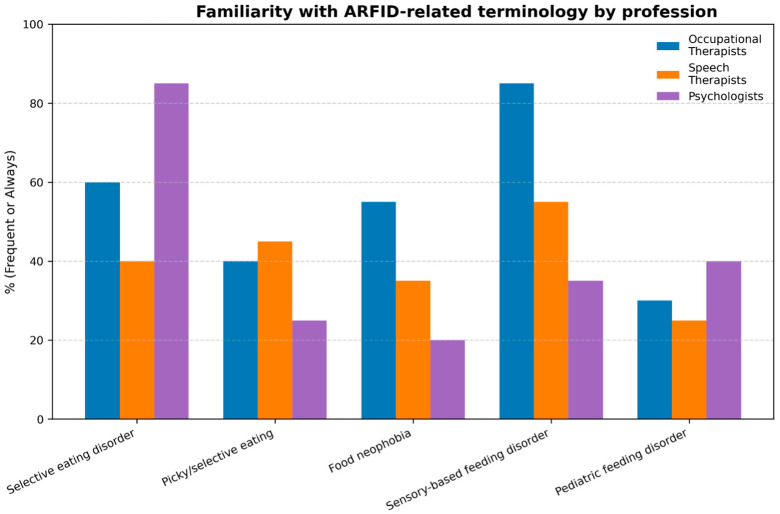
Frequency with which different professional profiles have heard or used terms referring to ARFID, corresponding to questionnaire Item 7.

**Figure 2 nutrients-17-03636-f002:**
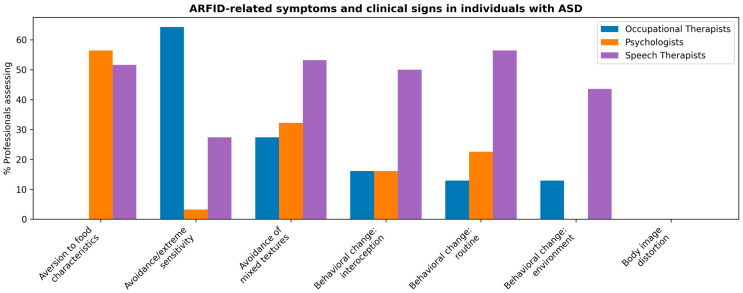
Symptoms and signs indicative of ARFID as considered according to professional profile, corresponding to questionnaire Item 9.

**Figure 3 nutrients-17-03636-f003:**
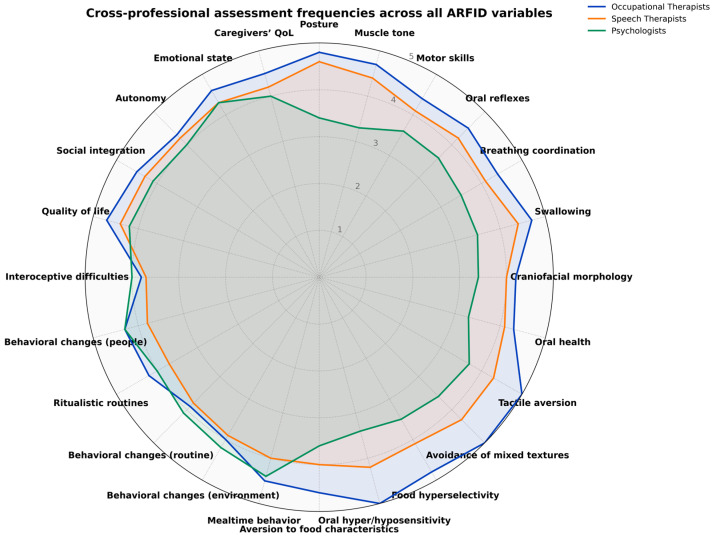
Radar chart of assessment areas and professional profiles involved in the evaluation and diagnosis of ARFID in individuals with ASD. The figure integrates items from the clinical evaluation section of the questionnaire: swallowing safety and efficiency (Items 12a–12f), sensory aspects (Items 13a–13e), behavioural factors (Items 14a–14f), and psychosocial aspects (Items 15a–15e). Each axis corresponds directly to a single questionnaire item, and values reflect descriptive frequency data rather than psychometric domain scores.

## Data Availability

The data presented in this study are not publicly available due to privacy and ethical restrictions. Anonymized datasets may be made available from the corresponding author upon reasonable request and subject to approval by the Ethics Committee of the University of Castilla-La Mancha.

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
