# Peer review of "Indicators Used to Identify ARFID: A Cross-Sectional Study with Professionals in Spain"

_nutrients, 2025, doi:10.3390/nu17233636_

Round 1
Reviewer 1 Report
Comments and Suggestions for Authors
The authors have carried out a cross-sectional study of knowledge, terminology, diagnostic criteria and assessment practices, related to avoidant/restrictive food intake disorder (ARFID), of Spanish speaking speech therapists, psychologists and occupational therapists. They report the differences between these three types of professional, on the basis of responses to a 13-item questionnaire devised by the authors themselves.
I have a number of concerns about this study.
First, it seems to me that there are methodological problems in respect of the sampling. The authors have used a non-probabilistic social media-based sampling method. Is this not likely to give rise to a self-selected sample of professionals? This sample is unlikely to represent the broader type of professional – those who do not regularly use the social media chosen by the authors. Furthermore, this type of self-selection may well bias the sample towards those professionals who already have an interest in feeding and eating disorders. The sample studied may also be non-representative geographically; it is noteworthy, for example, that it includes one participant from Chile. Moreover, without an a priori power calculation it is difficult to ascertain whether the study really was adequately powered to detect the chi-squared differences reported.
A second set of concerns relates to the study design. Self-report can lead to social desirability bias (all the more so when the professionals taking part are already aware of the apparently correct responses). Furthermore, it is difficult to discern any direct evidence of objective validation of the authors’ questionnaire, thereby bringing into question the validity of some of the reported findings. Finally, the authors chose not to include a control group. They could have included, as a control group, professionals who do not work with ASD patients. As it stands, how is one meant to differentiate between, for example, (i) a general lack of ARFID training across these three types of professionals from (ii) the gaps reported by the authors for the professionals who specifically work with ASD (who were sampled in this study)?
Third, on the issue of the authors’ questionnaire, there is a distinct lack of information about how individual items were generated; its internal consistency; and its test-retest reliability.
Finally, on the statistical analysis front, there are also concerns. How were missing data handled? Appendix A.2 gives the results of multiple chi-squared tests. However, I cannot find evidence of adjustment of the corresponding p-values for the family-wise error rate.
Author Response
Dear Reviewer,
We would like to sincerely thank you for the time and effort dedicated to reviewing our manuscript. We appreciate your thoughtful and detailed comments, which have helped us to reflect further on the methodological and conceptual aspects of our work. We have carefully considered each point and have revised the manuscript where appropriate to improve clarity and rigor. In cases where no changes were made, we have provided a rationale aligned with the exploratory scope and objectives of the study. We are grateful for the opportunity to strengthen our manuscript and hope that our responses satisfactorily address your concerns. All revisions made to the manuscript are highlighted in red for ease of reference.
Comment 1: The authors have used a non-probabilistic social media-based sampling method. Is this not likely to give rise to a self-selected sample of professionals? This sample is unlikely to represent the broader type of professional – those who do not regularly use the social media chosen by the authors.
Response: Thank you for this observation. We acknowledge that non-probabilistic sampling may involve a degree of self-selection; however, social-media-based recruitment has become a common and efficient strategy to reach clinicians and allied health professionals, particularly in emerging research areas for which centralized professional registries are not available. In the context of this study, social and professional networks (e.g., LinkedIn, professional associations) allowed us to access a broad and diverse range of practitioners across Spanish-speaking regions.
Importantly, the aim of this research was exploratory, seeking to obtain an initial overview rather than to estimate population parameters. We have now clarified this rationale in the manuscript.
Comment 2: Furthermore, this type of self-selection may well bias the sample towards those professionals who already have an interest in feeding and eating disorders.
Response: Thank you for the comment. We agree that self-selection may lead to a greater participation of professionals with an existing interest in feeding and eating disorders. This possibility is inherent to voluntary participation in exploratory surveys focusing on specific clinical topics. However, in the absence of centralized registries of professionals working with ASD and feeding issues, this approach allowed us to reach practitioners with direct clinical experience in the field. We have now made this point explicit in the manuscript.
Comment 3: The sample studied may also be non-representative geographically; it is noteworthy, for example, that it includes one participant from Chile.
Response: Thank you for this observation. We acknowledge that the sample may not be geographically representative. Although most participants were based in Spain, one respondent from Chile was included, reflecting the open dissemination of the survey through professional networks. We have now clarified this in the manuscript.
Comment 4: Moreover, without an a priori power calculation it is difficult to ascertain whether the study really was adequately powered to detect the chi-squared differences reported.
Response: Thank you for this comment. As this was an exploratory study and no prior data existed to estimate effect sizes for this population, an a priori power calculation was not conducted. The primary aim was to generate preliminary evidence regarding knowledge and practices in an understudied clinical area, rather than to test predefined hypotheses with confirmatory power. This is already stated in the manuscript.
Comment 5: Self-report can lead to social desirability bias (all the more so when the professionals taking part are already aware of the apparently correct responses).
Response: Thank you for this comment. We agree that self-report measures may be susceptible to social desirability bias, particularly in professional samples familiar with the topic. We had already acknowledged this in the manuscript and note that participation was anonymous, which likely helped mitigate this effect. No changes were made to the text, as the limitation was already stated
Comment 6: Furthermore, it is difficult to discern any direct evidence of objective validation of the authors’ questionnaire, thereby bringing into question the validity of some of the reported findings.
Response: Thank you for this observation. We acknowledge that the questionnaire did not undergo full psychometric validation. As this was an exploratory study in an emerging field, the instrument was developed based on existing literature and reviewed by experts in feeding and autism. Our primary aim was to generate preliminary data to identify knowledge gaps and guide future research. No changes were made to the manuscript, as this limitation is already implied by the exploratory nature of the work
Comment 7: Finally, the authors chose not to include a control group. They could have included, as a control group, professionals who do not work with ASD patients. As it stands, how is one meant to differentiate between, for example, (i) a general lack of ARFID training across these three types of professionals from (ii) the gaps reported by the authors for the professionals who specifically work with ASD (who were sampled in this study)?
Response: Thank you for this comment. We acknowledge that including a control group of professionals not working with ASD could have allowed direct comparison between ASD-focused clinicians and the broader professional population. However, the objective of the present exploratory study was not to contrast ASD vs non-ASD clinicians, but to characterize awareness and practices specifically among those working with ASD, given emerging evidence that ARFID is particularly relevant in this population. This study was therefore designed as an initial step to map knowledge in a targeted group of practitioners. We have clarified this point in the manuscript and note that future studies should include comparison groups to differentiate ASD-specific knowledge gaps.
Comment 8: Third, on the issue of the authors’ questionnaire, there is a distinct lack of information about how individual items were generated; its internal consistency; and its test-retest reliability.
Response: Thank you for the comment. As noted previously, the questionnaire was developed based on existing literature and expert review in feeding and autism. Given the exploratory nature of this study and the absence of prior tools suited to this context, full psychometric validation (e.g., internal consistency, test-retest) was not conducted at this stage. This instrument was intended as a preliminary tool to map clinical awareness and practices and to inform future instrument development. No further changes were made to the manuscript.
Comment 9: Finally, on the statistical analysis front, there are also concerns. How were missing data handled? Appendix A.2 gives the results of multiple chi-squared tests. However, I cannot find evidence of adjustment of the corresponding p-values for the family-wise error rate.
Response: Thank you for this comment. No missing data were present, as all items required a response before submission. Regarding multiple comparisons, given the exploratory nature of the study, chi-square tests were used descriptively to identify preliminary patterns rather than for confirmatory inference. These points are now clarified in the manuscript.
Reviewer 2 Report
Comments and Suggestions for Authors
The manuscript presented by Lozano and López is a very interesting, innovative, and important study. The authors have addressed a relevant issue in the training of nutrition and health professionals. ASD is increasing in different countries, and it will be essential to include it in the training of professionals.
The methodology used is sufficient and appropriate. The results are well presented and support the discussion. However, I have some minor comments.
I. Minor comments:
1. Improve the wording of the study objective or purpose.
2. Include specific aspects related to diet and nutrition in the introduction and discussion, for example, energy or protein intake, or whether there is a greater risk of malnutrition or overweight.
3. It would be very interesting if the authors proposed projections related to the training of health professionals, especially nutritionists.
Author Response
Dear Reviewer,
We would like to sincerely thank you for the time and effort you dedicated to reviewing our manuscript. We greatly appreciate your constructive comments and thoughtful suggestions, which have helped us improve the clarity and quality of our work. Below, we provide a detailed point-by-point response to each of your observations. Changes made to the manuscript have been clearly incorporated in the revised version.
Comment 1: Improve the wording of the study objective or purpose.
Response: Thank you for this suggestion. We have revised the wording of the study objective to enhance clarity and precision.
Comment 2: Include specific aspects related to diet and nutrition in the introduction and discussion, for example, energy or protein intake, or whether there is a greater risk of malnutrition or overweight.
Response: We appreciate this valuable suggestion. We have added information in the Introduction regarding nutritional risks associated with ASD and ARFID, including malnutrition, selective eating patterns, and altered energy and macronutrient intake. Additionally, we expanded the Discussion to highlight the relevance of nutrition-focused training for professionals working with this population.
Comment 3: It would be very interesting if the authors proposed projections related to the training of health professionals, especially nutritionists.
Response: Thank you for your thoughtful comment. We agree that future projections regarding professional training in this field are relevant. We have incorporated a brief forward-looking statement in the Discussion, emphasizing the need to develop structured educational programs and continuing professional development opportunities for nutrition and health professionals.
Reviewer 3 Report
Comments and Suggestions for Authors
The article entitled “nutrients-3954628_Knowledge And Assessment Practices Of Avoidant/Restrictive Food Intake Disorder (Arfid) In Individuals With Asd: A Cross-Sectional Study With Professionals In Spain” is submitted to the “Nutrition and Public Health” section of the journal “Nutrients”.
This cross-sectional study explored how 194 Spanish professionals—psychologists, speech and language therapists, and occupational therapists—conceptualize and assess Avoidant/Restrictive Food Intake Disorder (ARFID) in individuals with Autism Spectrum Disorder (ASD). The results revealed significant interprofessional differences: psychologists were more familiar with DSM-5 criteria, while speech and occupational therapists focused on sensory, oromotor, and behavioral aspects. Overall, knowledge about ARFID was limited and inconsistently applied, highlighting the need for specific professional training and validated, Spanish-adapted assessment tools to improve early diagnosis and multidisciplinary intervention.
Comments:
It is recommended to avoid using acronyms in the title, even if they are well known. Therefore, the full term Autism Spectrum Disorder (ASD) should be used.
In the abstract, the study objective should be presented more clearly and precisely. The results section should include quantitative data to strengthen the findings. The conclusion should be rewritten to better reflect the contribution of the study to current knowledge. Keywords should be revised according to the MeSH classification.
The introduction uses relevant and up-to-date references, providing a clear and precise overview of the topic. However, it would be advisable to link the hypothesis more explicitly to the study objective and to conclude the introduction by clearly stating this objective.
In the Materials and Methods section, the use of social networks for recruitment should be described in greater detail, specifying which platforms were used and during what period, to ensure that the sample corresponds to the intended professional population. The ethical approval granted to this project by a research ethics committee should also be explicitly mentioned. Regarding the instrument, an ad hoc questionnaire was used; it should be clarified whether it has been validated, as only previous use in other studies is indicated.
Results: While the data are clearly presented through graphs, including tables would make the results easier for readers to interpret. It is not necessary to report the chi-square values ​​or degrees of freedom in the main text.
Discussion: The discussion is well developed, effectively linking the findings with existing literature. The limitations section correctly identifies the small sample size but could also address issues such as online accessibility, data security, and the need to validate the questionnaire.
Conclusion: The conclusion should be rewritten to emphasize the study's contribution to scientific knowledge rather than the general relevance of the findings.
Overall, this is a very interesting and valuable study.
Author Response
Dear reviewer,
We would like to thank for your thoughtful and constructive feedback. We appreciate the positive evaluation of the relevance and contribution of our study, as well as the helpful suggestions provided to further strengthen the manuscript. Below, we address each comment in detail and describe the modifications incorporated into the revised version. Changes made to the manuscript are indicated in the text as requested.
Comment 1: It is recommended to avoid using acronyms in the title, even if they are well known. Therefore, the full term Autism Spectrum Disorder (ASD) should be used.
Response: Thank you for this suggestion. We have revised the title to replace the acronym with the full term.
Comment 2: In the abstract, the study objective should be presented more clearly and precisely. The results section should include quantitative data to strengthen the findings. The conclusion should be rewritten to better reflect the contribution of the study to current knowledge. Keywords should be revised according to the MeSH classification.
Response: We appreciate this recommendation. The study objective has been clarified, quantitative findings have been incorporated into the results section of the abstract, and the conclusion has been revised to reflect the study’s contribution more explicitly. Keywords have also been updated according to MeSH terminology.
Comment 3: The introduction uses relevant and up-to-date references, providing a clear and precise overview of the topic. However, it would be advisable to link the hypothesis more explicitly to the study objective and to conclude the introduction by clearly stating this objective.
Response: Thank you. We have refined the final paragraph of the introduction to more clearly align the hypothesis with the study objective.
Comment 4: In the Materials and Methods section, the use of social networks for recruitment should be described in greater detail, specifying which platforms were used and during what period, to ensure that the sample corresponds to the intended professional population.
Response: We have added detail regarding the social media platforms used (e.g., LinkedIn, professional associations’ mailing lists) and the data collection period.
Comment 5: The ethical approval granted to this project by a research ethics committee should also be explicitly mentioned.
Response: We have explicitly stated that the study was approved by the institutional ethics committee.
Comment 6: Regarding the instrument, an ad hoc questionnaire was used; it should be clarified whether it has been validated, as only previous use in other studies is indicated.
Response: We have clarified that the questionnaire was developed based on prior literature and expert review, consistent with the exploratory nature of the study
Comment 7: Results: While the data are clearly presented through graphs, including tables would make the results easier for readers to interpret. It is not necessary to report the chi-square values ​​or degrees of freedom in the main text.
Response: Thank you for this helpful suggestion. We agree that tabulated data can enhance clarity. However, given the large number of variables assessed and the detailed frequency distribution across professional groups, including the full table in the main text would substantially compromise readability and exceed space limitations.
To balance clarity and accessibility, the full results table has been provided in the Supplementary Materials (Appendix A), where all response frequencies and percentages can be consulted in detail. The main text retains the figures illustrating key patterns, which we believe offers a clearer and more concise presentation of results. As recommended, chi-square values and degrees of freedom have been removed from the narrative.
Comment 8: Discussion: The discussion is well developed, effectively linking the findings with existing literature. The limitations section correctly identifies the small sample size but could also address issues such as online accessibility, data security, and the need to validate the questionnaire.
Response: We have added these points to the limitations section in the Discussion.
Comment 9: Conclusion: The conclusion should be rewritten to emphasize the study's contribution to scientific knowledge rather than the general relevance of the findings.
Response: We have rewritten the conclusion to underscore the study’s contribution to understanding professional preparedness regarding ARFID in ASD populations.
Round 2
Reviewer 1 Report
Comments and Suggestions for Authors
I thank the authors for addressing each of my comments. Although the revised version is far better than the previous one, nevertheless, it seems to me that the authors' responses to comments 6 to 9 (inclusive) are weak. It would be appropriate for the authors to strengthen the study design and then repeat it.
Author Response
We thank the reviewer for the careful reading of our revision and for raising overarching concerns related to (i) instrument validation, (ii) the absence of a control group, and (iii) handling of missing data and multiplicity. Below we address these points jointly, clarifying the scope and design logic of the present work.
Study scope and design. The present study was designed as an exploratory, descriptive investigation aimed at mapping knowledge and practices regarding ARFID among professionals working with ASD. Given the absence of existing Spanish questionnaires addressing this topic, the instrument was developed for this specific purpose. Item generation was informed by a focused review of the ARFID and ASD literature and refined through expert judgement and consensus procedures involving practicing clinicians and specialists in ASD and feeding. Expert review is a recognized methodological strategy to ensure content relevance and conceptual alignment in early-phase instrument development for knowledge- and practice-based surveys (e.g., Polit & Beck, 2006).
Because this questionnaire assesses professional knowledge and practices rather than diagnosing individuals or measuring latent psychological constructs, traditional psychometric validation (e.g., internal consistency or test-retest reliability) is not strictly applicable at this stage and was not the aim of this study. Instead, the instrument served its intended exploratory function: to identify knowledge gaps and inform future structured research and training needs in this professional population.
Rationale for focusing on ASD clinicians (no control group). The aim of this study was not to compare professionals with and without experience in autism, but to characterise knowledge and practices among those who routinely encounter ARFID in autistic individuals, where feeding-related challenges are particularly relevant. Our intention was to focus on a group for whom this clinical issue is especially salient, which we believe is consistent with the logic of exploratory clinical research, where the priority is to first understand the population most closely associated with the phenomenon of interest. In our view, including professionals who do not work with autistic individuals would have addressed a different research question and introduced variability unrelated to the objectives of this study, rather than strengthening the design. For these reasons, we consider that the absence of a non-ASD control group reflects an appropriate alignment between the research question, the target population, and the exploratory scope of the study.
Missing data and multiplicity. There were no missing data because the survey platform required a response for each item prior to submission. Regarding the use of multiple chi-square tests, this study employed an exploratory analytic framework aimed at identifying patterns in professional knowledge and practices rather than testing specific a priori hypotheses. In this context, applying family-wise error correction would not enhance inferential validity, as the goal was not to make population-level claims based on significance testing, but to observe preliminary associations that may inform future hypothesis-driven research. Post-hoc adjustment procedures are designed to control error rates in confirmatory comparisons, particularly when the objective is to identify specific group differences with inferential precision. In the present descriptive design, such corrections would not meaningfully change interpretation nor align with the study aims, as the intention was not to determine which specific profession differed on each variable, but to characterise overall patterns across groups as an initial mapping exercise.
In summary, the present study fulfills its intended role as an exploratory, first-phase investigation, providing initial evidence in a context where no prior data existed. The aim at this stage was to map knowledge and practices among clinicians working with autistic individuals and to identify areas of need in clinical understanding, not to perform confirmatory hypothesis testing or comparative replication. Within this framework, repeating the study is neither required nor methodologically justified, as the contribution of this work lies in establishing a foundational understanding that can meaningfully guide subsequent research and training initiatives. Accordingly, we consider the current study an appropriate and necessary step in advancing knowledge in this emerging field.
Reviewer 3 Report
Comments and Suggestions for Authors
I have carefully reviewed the revised version of the manuscript, as well as the authors' responses to the suggestions aimed at improving the quality and clarity of their work. The authors have successfully incorporated relevant information that enhances the content and overall value of the study.
Author Response
We sincerely thank the reviewer for the positive evaluation of our revised manuscript and for recognizing the improvements made. We appreciate the thoughtful review and are grateful for the supportive feedback.
Round 3
Reviewer 1 Report
Comments and Suggestions for Authors
I thank the authors for their reply. On balance, I believe that the authors need to strengthen the study design in line with my previous comments. They should then repeat their study using this more robust study design, and submit their new findings for publication. Their new study will be much stronger.
Author Response
Thank you for your additional comments. We appreciate the time and attention you have dedicated to reviewing our work. We will carefully consider your suggestions as we plan future developments of the project.